# Visual and Acoustic Aspects of Face Masks Affect Speech Intelligibility in Listeners with Different Hearing Statuses

**DOI:** 10.3390/audiolres15010007

**Published:** 2025-01-21

**Authors:** Pauline Rohner, Rasmus Sönnichsen, Sabine Hochmuth, Andreas Radeloff

**Affiliations:** 1Department of Otolaryngology, Head and Neck Surgery, University of Oldenburg, Steinweg 13-17, 26122 Oldenburg, Germany; 2Research Center Neurosensory Science, University of Oldenburg, Küpkersweg 74, 26129 Oldenburg, Germany; 3Cluster of Excellence “Hearing 4 All”, University of Oldenburg, Carl-von-Ossietzky-Str 9-11, 26111 Oldenburg, Germany

**Keywords:** audiovisual speech intelligibility, hearing aids, COVID-19, face mask, speechreading

## Abstract

*Background:* When speaking while wearing a face mask, sound transmission is attenuated, and visual cues are lost due to the covered facial movements of the speaker. In this study, we investigated the extent to which different face masks alter speech intelligibility in individuals with different degrees of hearing impairment. *Methods:* A total of fifty participants were divided into four hearing status groups according to the degree of hearing loss: normal levels (16), mild (13), moderate (11), and severe (10). A modified version of the Audiovisual German Matrix Sentence Test (AV-OLSA) was used to assess speech perception in noise in five conditions (audiovisual, audio-only, visual-only, surgical mask, and FFP2 mask). *Results:* Our results show that acoustic attenuations of face masks cause a small but similar decrease in speech reception thresholds (SRTs) in listeners of different hearing statuses. The effect of visual cues (visual benefit) on SRTs was stronger than the effect of acoustic attenuation but also did not differ significantly between the different hearing status groups, with a median difference of 1.5 dB for mild hearing loss, 2.9 dB for moderate hearing loss, and 2.7 dB for severe hearing loss. The best-aided hearing status did not correlate with visual benefit. *Conclusions:* Our research confirms the importance of providing visual cues for speech reception in noisy environments, especially for individuals with impaired hearing, regardless of their degree of hearing loss.

## 1. Introduction

About 20% of the world’s population (more than 1.5 billion people) is affected by hearing loss. Causes of hearing loss include biological (e.g., diseases, genetics), environmental (e.g., noise exposure), and behavioral factors (e.g., smoking) [1]. Adverse listening situations involving noisy environments worsen the impact of hearing impairment on daily communication [2]. In addition to acoustic information, visual cues are crucial for sufficient speech understanding. The COVID-19 pandemic and the associated wearing of face masks in daily life highlighted the importance of both visual and acoustic cues for communication. However, the process of understanding speech does not only include the addition of acoustic and visual information but also the integration of both in the context of the individual knowledge and the individual memory process [3].

Recent research in light of the COVID-19 pandemic has further advanced knowledge of audiovisual speech understanding, especially when face masks are involved. Several studies have investigated the effect of face masks on communication, ranging from investigations of the acoustic attenuation properties of different mask types [4] to speech perception without visual aspects [5,6,7] and audiovisual speech perception [3,8,9,10]. A recent review identified 15 relevant publications on the effect of face masks on verbal communication, highlighting the influence of visual cues, acoustic attenuation, background noise, and hearing ability [11]. Some of these studies showed that the absence of visual cues due to the use of face masks can potentially reduce speech intelligibility. Different listener profiles have been studied, including normal hearing profiles, impaired hearing without compensation, hearing aids (HAs), cochlear implants (CIs), and auditory brainstem implant (ABI) listeners, with varying results. Some studies investigating the visual impact of face masks suggest that the lack of visual input leads to a significant reduction in speech intelligibility even in individuals with normal hearing levels [8,12], whereas other studies have found such changes only in individuals with impaired hearing [10,13,14,15]. However, comparisons between studies are somewhat challenging due to the different methodologies used to measure speech perception performance [11]. Most studies have assessed speech perception at a fixed presentation level in silence or in noise at a fixed signal-to-noise ratio (SNR). These methods tend to introduce ceiling effects, particularly in individuals with normal hearing or mildly impaired hearing, potentially obscuring potential effects within the results.

Some research has suggested a possible relationship between hearing ability and the benefit of available visual cues for speech perception in noise. Atcherson et al. 2017, for instance, found a significant improvement in speech intelligibility with noise when visual cues were available, but only for a group of listeners with severe hearing impairments—not for those with normal hearing levels or moderate hearing impairments. However, this study had shortcomings, including ceiling effects in the group with normal hearing levels and small, very diverse groups of hearing-impaired listeners [10]. In contrast, Sönnichsen et al. 2022b circumvented the ceiling effect for groups with good speech intelligibility by measuring speech reception thresholds at 80% intelligibility in noise. They found that CI users who solely listened with their implants benefited most from additional visual cues compared to individuals with normal hearing levels, single-sided deafness CI users, or CI users with asymmetric hearing loss [9]. It can, therefore, be hypothesized that listeners with more severe hearing loss rely more heavily on speechreading to compensate for their reduced auditory capacity compared to those with normal hearing levels.

In order to further our understanding of the potential relationships between the degree of hearing loss and intelligibility gained from the presence of visual cues, this study attempts a more systematic approach. The approach was to examine different degrees of hearing loss, specifically including normal hearing levels, mildly impaired hearing, moderately impaired hearing, and severely impaired hearing, according to the former World Health Organization criteria for hearing loss levels. We selected participants with only symmetric sensorineural hearing loss, some of whom used hearing aids to compensate. In addition, the experimental setup consisted of a thoroughly evaluated speech-in-noise test—the female version of the German matrix sentences test [16,17]—with a previously evaluated visual extension [18,19]. In addition to visual cues, the acoustic attenuation effects of two common masks (surgical and FFP2) were investigated.

## 2. Materials and Methods

This study was conducted at a tertiary referral center. Participants with normal hearing levels and impaired hearing were recruited from the outpatient clinic and hospital ward, and all gave written informed consent. The study was approved by the local ethics committee (AZ 2020-135).

### 2.1. Participants

A total of 50 participants were selected and categorized into four groups based on their hearing statuses, following the World Health Organization’s grades of hearing impairment. This classification was determined by the pure tone average at 0.5, 1, 2, and 4 kHz (PTA4) of the better ear: it included normal hearing levels (PTA4 ≤ 25 dB HL), mild hearing loss (PTA4: 26–40 dB HL), moderate hearing loss (PTA4: 41–60 dB HL), and severe hearing loss (PTA4: 61–80 dB HL). Average pure tone thresholds are shown in Figure 1. Participants with hearing impairment (HI) had symmetric sensorineural hearing loss without significant conductive components. Symmetric hearing loss was defined as a maximum PTA4 difference of 15 dB HL or less between the ears. The tests were performed with their personal hearing aids if used in daily life (see Table 1 for details). In this case, aided thresholds were also obtained. All participants were at least 18 years old, had self-reported normal or corrected-to-normal vision, no cognitive impairment, and were native German speakers. Demographic data are shown in Table 1.

Although we aimed to have age-matched groups, there was a statistically significant difference in age between the four hearing status groups, as revealed by the Kruskal–Wallis test (H(3) = 13.73, *p* = 0.003). Bonferroni-corrected pairwise comparisons indicated that moderate and severe HI groups were significantly older than the group of NH listeners (*p =* 0.013 and *p =* 0.014, respectively), while other differences were not significant.

### 2.2. Experimental Design

Measurements were performed in a sound-treated room. Each participant was seated at a distance of 80 cm from a loudspeaker (8030C studio monitor, Genelec, Isalmi, Finland) and a 23.8″ monitor (P2419H, DELL GmbH, Frankfurt, Germany). An audiovisual version of the Female German Matrix Test was used [18,19]. Speech intelligibility was measured in the test-specific masking noise in five separate test conditions in random order:-Audiovisual (AV) test;-Audio-only (AO) test;-Audiovisual with a surgical mask type II—EN14683 (AVM-S);-Audiovisual with FFP2 mask type NR—EN149:2001 + A1:2009 (AVM-FFP);-Visual-only (VO) test.

In the two mask conditions, a mask-shaped object was inserted into the video material to cover the mouth and nose of the speaker, and the audio signal was attenuated according to the respective mask. The same attenuated signals used by Sönnichsen et al. [9] were used in this study. Attenuation was performed using a filter derived from recordings of four different speakers speaking matrix sentences with and without a respective mask. The largest attenuation effects occurred in the frequency range from 2.5 to 8 kHz: about 4 dB for the surgical mask and about 6.5 dB for the FFP2 mask. The filter addressed the frequency-specific attenuation effects of the masks; absolute loudness effects were avoided by equating the RMS level of the sentences. The frequency spectra of the attenuation properties and a more detailed description of the signal generation can be found in the work of Sönnichsen et al. [9]. In the audio-only and visual-only conditions, participants were exposed solely to unattenuated speech and noise and video and noise, respectively.

A different, randomly selected test list of 20 sentences of the audiovisual Female Matrix Test was used for each condition. Participants were instructed to repeat the words they recognized and were encouraged to guess if they were unsure. Except for the visual-only condition, the speech reception thresholds at 80% intelligibility (SRT_80_) were determined using an adaptive procedure in which the speech level was adjusted based on the participants’ responses while the background noise level was kept constant at 65 dB SPL. The initial SNR of the adaptive procedure was set to −5 dB, and the procedure was limited to a maximum SNR of 20 dB. See Llorach et al. [18] for a detailed description of the adaptive procedure used. In the visual-only condition, performance was measured based on the percentage of correctly repeated words. To reduce possible training effects, two test lists in the audiovisual condition were applied beforehand.

Acoustic signals were calibrated to a level of 85 dB SPL using a PCE 322A c-weighted level meter (PCE Deutschland GmbH, Meschede, Germany) placed at the participant’s head position. The video signal was synchronized with the audio signal using an external camera, as described by Llorach et al. [18].

### 2.3. Analysis

All data were analyzed using IBM SPSS Statistics (version: 29.0.0.0 (241)), and graphs were created using GraphPad Prism (version 9.5.1). A *p*-value of less than 0.05 was considered statistically significant. Non-parametric statistical tests were used due to the small sample size and varying group sizes. The Kruskal–Wallis test was used for comparisons between groups. The Friedman test was used to test conditions within a hearing status group. Spearman’s rank sum test was used for correlation analysis. All data are expressed as medians and interquartile ranges (IQRs).

To differentiate between the visual and acoustic effects of face masks for speech intelligibility in the different hearing status groups, SRT differences were analyzed. A visual benefit was defined as the difference between the SRT_80_ in the AV condition and the SRT_80_ in the AO condition. The influence of the mask in terms of access to visual information can be seen both as a loss of visual cues due to the mask covering the mouth and nose area and conversely as a visual benefit when the mask is removed and visual information is added. In the following sections, we used the term visual benefit rather than the loss of visual cues. The acoustic attenuation caused by the two mask types was calculated as the difference between SRT80 in the AVM-S/AVM-FFP conditions and SRT80 in the AO condition.

## 3. Results

### 3.1. Speech Reception Thresholds

Within each hearing status group, the SRT80 was lowest in the AV condition and highest in the condition with the FFP mask (Figure 2). SRT80s generally increased from the group with normal hearing levels with the lowest SRT80s to the group with severe hearing impairment with the highest SRT80s, although the number of aided measurements increased with increasing hearing loss. Median SRT80s and interquartile ranges are shown in Table 2 for each condition and hearing status group, including the training data.

Statistical comparisons across conditions and training data were performed separately for each hearing status group using the Friedman test, which was significant in each group (normal levels: χ^2^(5) = 65.197, *p* < 0.001; mild: χ^2^(5) = 45.353, *p* < 0.001; moderate: χ^2^(5) = 42.552, *p* < 0.001; severe: χ^2^(5) = 29.655, *p* < 0.001). Eight pairwise comparisons were evaluated, and significant values were corrected, respectively, using Bonferroni correction. The eight pairwise comparisons included AV condition with AO, AVM-S, AVM-FFP, the first and second training AV measurement, AO and the two masked conditions, and the two masked conditions. Adjusted significant differences are shown in Figure 2 (except for training data). A possible training effect was minimized by training, as there was clearly no significant difference between the SRT80 of the second training measurement and the AV measurement (normal levels: *p* = 2.758, mild: *p* = 1.824, moderate: *p* = 5.860, severe: *p =* 0.754). A significant visual benefit was found in all groups except for the listers with mild hearing impairment. Acoustic attenuation alone was significant only in the group with normal hearing levels and in the group with mildly impaired hearing for the FFP mask. When visual loss and acoustic attenuation caused by the FFP mask were combined, the largest and most significant effect on SRT80 was observed in each group.

To compare the visual benefit and acoustic attenuation effects between hearing status groups, further analyses were conducted on SRT differences. A visual benefit was calculated as the SRT80 difference between the AV and AO conditions; acoustic attenuation was calculated as the SRT80 difference between the respective masked AVM-S/AVM-FFP condition and the AV condition. Both the visual benefit and the two acoustic attenuations produced comparable improvements or deteriorations in SRT80 compared to the AO condition (Figure 3).

The visual benefit ranged from −2.9 dB (moderate HI group) to −1.5 dB (mild HI group), while the greatest reduction in SRT80 due to acoustic attenuation was observed in the FFP condition with values ranging from 1.5 dB (moderate HI group) to 2.6 dB (severe HI group); see Table 2 for medians and interquartile ranges of each hearing status group. Kruskal–Wallis tests, performed separately for the visual benefit and the two acoustic attenuations, confirmed the observed similar effects across hearing status groups (visual benefit: H(3) = 4.832, *p =* 0.185; surgical mask attenuation: H(3) = 3.883, *p =* 0.274; FFP mask attenuation: H(3) = 1.092, *p =* 0.779).

### 3.2. Speechreading Performance

Visual-only scores were obtained to assess the speechreading ability of NH and HI participants (only visual information was provided, no audio information). Median VO scores were 29% (14.8–36.3%) for the NH group, 19% (7–26%) for the mild HI group, 28% (20.5–41.5%) for the moderate HI group, and 8.5% (2–12.8%) for the severe HI group. Individual speechreading skills varied considerably between listeners within each group (Figure 4). The Kruskal–Wallis test performed on the VO scores showed no statistically significant effect on the hearing status factor (H(3) = 6.474, *p =* 0.091).

### 3.3. Impact on Visual Benefit

The influence of various aspects, including age, unaided pure-tone thresholds (PTA4), speechreading ability (VO), and speech perception in noise (AO) on visual benefit was analyzed using correlation analysis. Furthermore, the effects of these aspects on each other were investigated (Figure 5).

Hearing status in terms of the mean pure tone audiometry (PTA4) and mean best-aided pure tone audiometry (best-aided PTA4) of both ears was not correlated with visual benefit (*r* = −0.2, *p* = 0.17, n = 50 and *r* = −0.18, *p* = 0.22, n = 48, respectively), indicating no association between unaided and aided hearing status and visual benefit. A moderate correlation was observed between speechreading scores and visual benefit (*r* = −0.45, *p* = 0.001, n = 50), indicating a higher visual benefit when speechreading performance was better. No significant correlation was found between the speechreading scores and speech-in-noise perception (*r* = −0.14, *p* = 0.33, n = 50) and best-aided PTA4 (*r* = −0.12, *p* = 0.41, n = 48). However, age was significantly correlated with speechreading scores (*r* = −0.42, *p* = 0.003, n = 50, Figure 5), indicating that elderly individuals exhibited poorer performance in speechreading than younger individuals, but this effect did not translate to visual benefits, as age and visual benefit were not correlated (*r* = 0.27, *p* = 0.06, n = 50).

## 4. Discussion

This study investigated the impact of face masks on speech intelligibility in individuals with varying degrees of hearing loss, with a special focus on acoustic and visual cues. We hypothesized that people with a higher degree of hearing loss would be more affected by face masks because they are more dependent on visual cues and more sensitive to acoustic attenuation than those with normal hearing levels.

### 4.1. Acoustical Attenuation

Two masks with different attenuation characteristics were used in this study to investigate the effect these masks would have on speech intelligibility for listeners with different hearing statuses. We found that the acoustic attenuation of the surgical mask did not significantly affect speech intelligibility in any of the hearing status groups. However, the acoustic attenuation of an FFP2 mask resulted in a significant decrease in SRT values compared to the audio-only condition, but only in NH and mild HI listeners (Figure 2). Similarly to our findings, Rahne et al. 2021 [5] found a greater attenuation effect of an FFP2-type mask compared to a surgical mask, although both mask types significantly reduced intelligibility in NH listeners. In our study, the magnitude of attenuation did not differ between hearing status groups for either type of mask: when all groups were compared, there was no significant difference in the magnitude of acoustic attenuation for either mask (Figure 3). In contrast to our findings, Alkharabsheh et al., 2023 [20] showed that the acoustic effects of an N95 mask had a greater impact on HI listeners with mild-to-moderate sensorineural hearing loss compared to NH participants. However, this was determined using live speech and three levels of background noise (SNR +10 dB, +5 dB, 0 dB). Their results may differ since we used a different testing environment, the use of hearing aids in Alkharabsheh et al’s previous study remains unclear, and our cohort consisted of a wider range of hearing loss. Homans and Vroegop, 2021 [14] also used live speech for testing and found a negative acoustic effect of a transparent (speechreading available) face shield for HI listeners with poor speech understanding in quiet conditions. This face mask type, which is not included in our study, has a poorer acoustic performance compared to surgical and FFP2 masks [4]. It shows that for other mask types, acoustic attenuation could play a more significant role in speech intelligibility in HI listeners.

### 4.2. Visual Benefit

A significant improvement in speech understanding has been described when visual information is added to a speech signal for NH listeners and cochlear implant users [8,9]. However, when analyzing subgroups of cochlear implant users, Sönnichsen et al. found that the visual benefit was greater only in cochlear implant users who heard only with their CI compared to individuals with normal hearing levels. CI users with better residual hearing in the other ear (asymmetric hearing loss or single-sided deafness) did not have a significantly better visual benefit compared to listeners with normal hearing levels [9]. These findings suggest that there is a greater visual benefit for listeners with greater hearing impairment. In the current study, we were able to show a significant visual benefit for all hearing status groups, except for participants with mild hearing loss, possibly due to the small effect size (Figure 2). However, a significant difference was observed when comparing the audiovisual and surgical mask conditions in this group, reflecting a comparable effect to that when comparing AV and AO, which was defined as a visual benefit in our study (SRTs did not differ between AO and the surgical mask conditions in this group). Contrary to our hypothesis, there was no significant difference in visual benefit between the hearing status groups (Figure 3). Gieseler et al., 2020 [21] evaluated speech reception thresholds in noise through the AV-OLSA and assessed audiovisual integration abilities with McGurk illusion responses in both NH and mildly HI individuals without hearing aids. Their findings align with ours in that they describe no significant difference in visual benefit between mildly HI and NH individuals. Similarly, Tye-Murray et al., 2007, describe no differences in visual enhancement in consonant, word, and sentence tests in a cohort of individuals with mild-to-moderate hearing impairments and normal hearing levels [22]. On the other hand, Atcherson et al., 2017 [10] investigated the effect of a transparent and paper mask on speech perception. They included participants with normal hearing levels and moderate and severe-to-profound hearing loss and used stable background noise (SNR +10 dB) to test speech reception. Their study revealed that the most significant deterioration effects caused by the loss of visual signals occurred in the group with severe-to-profound hearing loss, implicating the largest visual benefit for that group.

### 4.3. Auditory–Visual Speech Recognition

Multiple mechanisms contribute to auditory–visual speech recognition, as elucidated by Grant, Walden, and Seitz, 1998 [3]. Particularly, the fusion of auditory and visual stimuli, the use of speechreading, and the encoding of auditory information play a crucial role in speech perception within complex communication environments. To further investigate the possible contributing factors that influence visual benefit, we analyzed the speechreading abilities of all participants. We found a significant correlation between visual benefit and speechreading scores (Figure 5), underscoring the importance of speechreading for audiovisual speech understanding. As has been described before, we were expecting greater speechreading scores in participants with severe hearing loss [23,24]. However, no significant differences in speechreading scores were observed between the hearing status groups (Figure 4). Similarly, speechreading abilities were not correlated with speech perception in noise or aided hearing level (audio-only and best-aided PTA4, respectively). In contrast, a study examining the influence of face masks on HI individuals with hearing devices found that speech perception in quiet conditions correlated with the impact of a surgical mask on speech intelligibility, indicating the more pronounced effect of masks on individuals exhibiting lower speech intelligibility [14]. However, in this study, the impact of the surgical mask was measured using a speech tracking test in quiet conditions with live speech, and the visual and acoustic components of face mask attenuation were not analyzed separately, making it difficult to compare these results to our findings.

It is important to note that all but one participant in our study had acquired hearing loss post-lingually, as better speechreading scores have been described in individuals with early-onset hearing impairment [23]. Individuals with early-onset hearing loss may have superior decoding strategies for visual information [25]. Conversely, it has been proposed that the use of hearing aids may even result in enhanced speechreading abilities [26]. This suggests a potential link between hearing aid use, improved cognitive functions, and speechreading abilities [27]. However, the current literature is inconsistent concerning the relationship between hearing loss and speechreading abilities [22,28], but it appears that advanced speechreaders are better able to utilize visual information and thereby demonstrate a greater visual benefit [3].

Cognitive functions, and thus, age, appear to influence audiovisual integration, as age-related declines in cognitive functions (e.g., working memory) have been reported [29,30]. We found a significant decrease in speechreading scores with increasing age (Figure 5). However, this factor seems to have an indirect impact on audiovisual integration via unimodal perceptual abilities, which has not been accounted for in this study [31]. In line with these findings, we did not find a correlation between age and visual benefits, which could also be attributed to possible superior verbal intelligence skills and word knowledge, which are known to improve with age [30]. The positive effects of hearing rehabilitation on cognition have been reported, particularly in the context of hearing aid use [27]. Rosemann et al., 2021 [32] investigated the impact of hearing aid use on audiovisual integration. They reported a reduction in resting state connectivity between the auditory cortex and fusiform gyrus in HI participants after six months of hearing aid usage, possibly resulting in altered audiovisual integration abilities. Notably, we included participants with varying durations of hearing aid usage (ranging from 6 months to 39 years) as well as participants without hearing aids. This could partially explain the broader range of visual benefits in HI listeners (ranging from −10.5 dB to 0 dB) compared to listeners with normal hearing levels (ranging from −4 dB to 0.2 dB) in our study.

### 4.4. Limitations

Generally, the age distribution was not equal in the groups of HI compared to NH individuals due to the higher prevalence of hearing impairment in the elderly. One could expect, as mentioned previously, that with increased age, there is a decline in cognitive skills, leading to poorer performance in working memory function [33]. Moreover, sensory deficiencies like vision loss may have contributed to our results since we did not control visual impairments and relied on self-reported unrestricted vision. The standardized assessment of the inclusion criteria “absence of vision loss” and “no cognitive impairment” should be included in further studies. The audio-only condition was used in this study to calculate the acoustic effects of face masks. We found this to be sufficient since our previous study found no difference in audio-only and masked audiovisual conditions with unattenuated audio cues in NH individuals [8]. However, the results of NH individuals cannot be directly transferred to individuals with impaired hearing, and possible effects have not been addressed in this study. The limited group size is another limitation of this study. Further studies should include a larger cohort with a wide range of hearing loss severity. In addition, measurements of audiovisual integration, such as McGurk illusions or functional MRI techniques, could be helpful to better understand the mechanisms behind visual benefit [21,34].

## 5. Conclusions

Our results provide evidence for the deteriorating effects of face masks on speech reception thresholds in individuals with and without hearing impairment. The superior factor contributing to this effect is the loss of visual cues. Our findings are relevant not only for clinicians engaged in caring for HI patients but also in many fields where effective communication with HI individuals is important, such as educational settings and public authorities. We did not find a direct link between hearing status groups or the individual (aided) hearing level and visual benefit. Future studies should include larger cohorts and focus on a multifactor approach when investigating the complex processes involved in understanding audiovisual speech.

## Figures and Tables

**Figure 1 audiolres-15-00007-f001:**
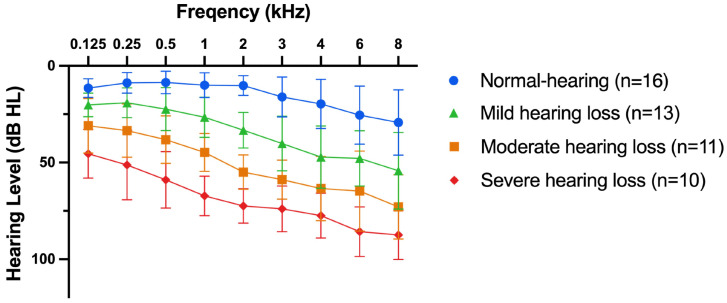
Mean pure tone thresholds of both ears for all participants (n = 50) are shown. Blue: normal hearing, levels; Green: mild hearing loss; Orange: moderate hearing loss; Red: severe hearing loss. Mean ± SD.

**Figure 2 audiolres-15-00007-f002:**
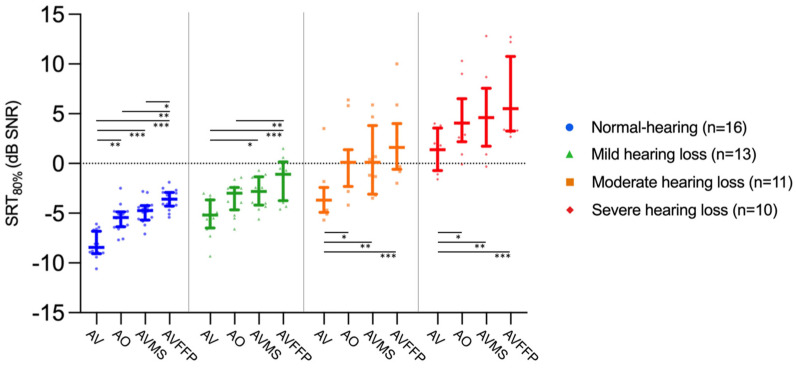
Speech reception thresholds (80%) for audiovisual (AV), audio-only (AO), surgical mask (AVM-S), and FFP2 mask (AVM-FFP) conditions are displayed for each hearing status group in sequence: normal hearing levels (blue), mildly impaired hearing (green), moderately impaired hearing (orange), and severely impaired hearing (red). Median values, interquartile ranges, and individual data are shown. Significant differences between conditions within each hearing status group are indicated by black horizontal lines with the corresponding level of significance shown as *** *p* < 0.001, ** *p* < 0.01, and * *p* < 0.05.

**Figure 3 audiolres-15-00007-f003:**
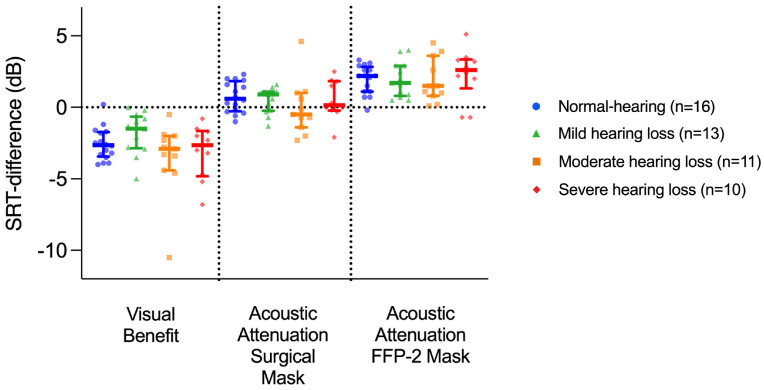
SRT_80_ differences in audiovisual and audio-only conditions (visual benefit) and audio-only and masked conditions (acoustic attenuation of a surgical mask and FFP2 mask) for different hearing status groups. Thick horizontal lines indicate mean values. Error bars indicate standard deviations. Dots show single data.

**Figure 4 audiolres-15-00007-f004:**
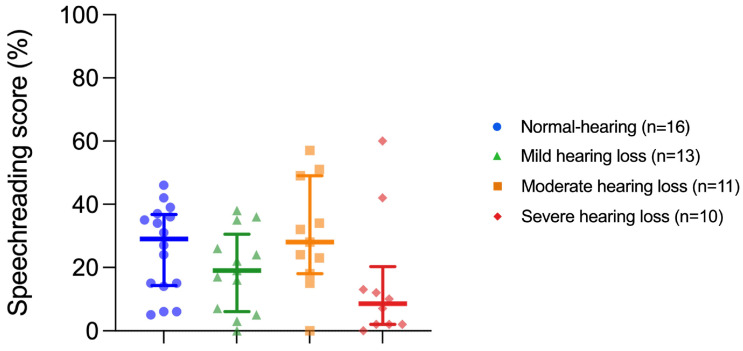
Speechreading scores (visual-only) for the hearing status groups (from left to right: normal hearing levels and mild, moderate, and severe hearing impairments). Medians, interquartile ranges, and single scores are shown.

**Figure 5 audiolres-15-00007-f005:**
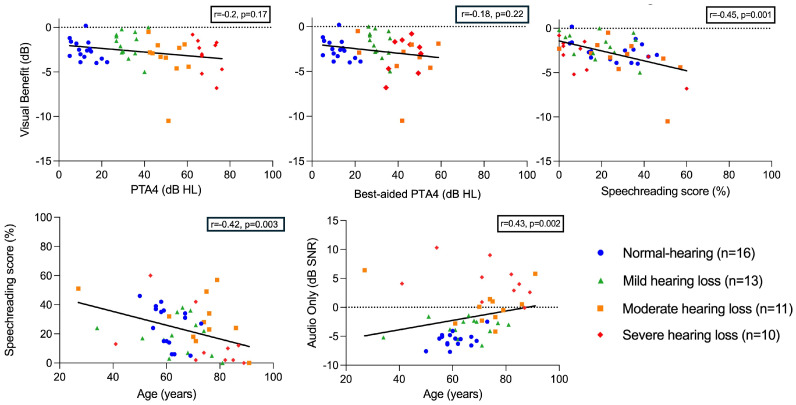
Scattered plots and correlation analysis for visual benefit (**upper panel**) and age (**lower panel**). Single scores and Spearman’s rank correlation values are shown.

**Table 1 audiolres-15-00007-t001:** Demographic data of all participants (n = 50) included in the study.

Category	Normal Hearing Levels (n = 16)	Mild Hearing Loss (n = 13)	Moderate Hearing Loss (n = 11)	Severe Hearing Loss (n = 10)
**Gender, amount (%)**				
Female	6 (38%)	3 (23%)	2 (18%)	5 (50%)
Male	10 (62%)	10 (77%)	9 (82%)	5 (50%)
**Age (years)** Median (IQR)	60.8 (±5.9)	64.4 (±12.1)	71.5 (±16.7)	73.7 (±15.5)
**PTA4 (dB HL)** Median (IQR)	12.2 (9.1–14.5)	30.0 (28.8–35.6)	50.0 (45.0–55.6)	66.9 (65.9–73.6)
**PTA4 best-aided (dB HL)** Median (IQR)	12.2 (9.1–14.5)	29.4 (26.9–35.6)	42.5 (39.4–50.0)	45.9 (39.4–49.8)
**Hearing aid (%)**	0 (0%)	2 (15%)	6 (55%)	10 (100%)
**Years of hearing loss** Median (IQR)	-	5 (2.5–10)	8 (4–10)	15 (10.5–19)
**Years hearing aid used** Median (IQR)	-	0 (0–0)	0.5 (0–7.5)	15 (8.5–19)

**Table 2 audiolres-15-00007-t002:** Median SRT80 and interquartile ranges for audiovisual (AV), audio-only (AO), surgical mask (AVM-S), and FFP2 mask (AVM-FFP) conditions and two training measurements in the AV condition are shown in the upper part. In the lower part, SRT differences between AV and AO (visual benefit) and between the two masked conditions AVM-S/AVM-FFP and AO (Atten. surgical/FFP) are shown as median and interquartile ranges.

	Normal	Mild	Moderate	Severe
	SRT80/dB SNR			
T1−AV	−5.3 (−6.4–−4.5)	−3.5 (−4.1–−2.0)	−0.6 (−1.8–3.5)	4.6 (2.7–5.9)
T2−AV	−7.3 (−8.5–−6.5)	−4.2 (−5.8–−3.9)	−2.4 (−4.5–−1.2)	1.7 (0.8–3.4)
AV	−8.5 (−9.0–−7.0)	−5.2 (−6.4–−3.8)	−3.7 (−4.8–−2.6)	1.4 (−0.3–3.1)
AO	−5.5 (−6.3–−5.0)	−3.0 (−4.1–−2.5)	0.1 (−2.0–1.2)	4.1 (2.7–5.6)
AVM−S	−4.8 (−5.6–−4.4)	−2.8 (−4.1–−2.5)	0.1 (−2.2–2.3)	4.6 (2.5–6.8)
AVM−FFP	−3.6 (−4.2–−2.9)	−1.1 (−3.6–0.1)	1.6 (−0.5–3.6)	5.5 (3.3–10.0)
	SRT difference/dB			
Visual Benefit	−2.7 (−3.4–−1.8)	−1.5 (−2.8–−0.8)	−2.9 (−3.9–−2.2)	−2.7 (−4.3–−1.8)
Atten. Surgical	0.6 (−0.2–1.7)	0.9 (−0.2–1.1)	−0.5 (−1.1–0.8)	0.2 (−0.2–1.7)
Atten. FFP	2.2 (1.1–2.7)	1.7 (0.9–2.9)	1.5 (0.9–3.1)	2.6 (2.1–3.3)

## Data Availability

The data presented in this study are available on request from the corresponding author (data are not publicly available due to privacy restrictions).

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
