# Peer review of "Visual and Acoustic Aspects of Face Masks Affect Speech Intelligibility in Listeners with Different Hearing Statuses"

_audiolres, 2025, doi:10.3390/audiolres15010007_

Round 1
Reviewer 1 Report
Comments and Suggestions for Authors
I read with extreme interest the paper and found it clear, well-written and easy to understand.
I suggest adding as a limitation of the article the fact that the groups of patients were limited. The great variations of people involved in the study i.e. type of support for hearing impairment) and the limitations described by the authors (age distribution, the absence of control for visual impairments and cognitive functions) imply that groups of 10-15 people are insufficient to generalize the results obtained. However, the results reported offer food for thoughs.
After this minor revision, I am favourable to the acceptance of the manuscript.
Author Response
Comment 1: I suggest adding as a limitation of the article the fact that the groups of patients were limited. The great variations of people involved in the study i.e. type of support for hearing impairment) and the limitations described by the authors (age distribution, the absence of control for visual impairments and cognitive functions) imply that groups of 10-15 people are insufficient to generalize the results obtained. However, the results reported offer food for thoughs.
Response 1: We appreciate your comment and edited the limitations section accordingly (p. 12, line 396).
Reviewer 2 Report
Comments and Suggestions for Authors
This study investigated the impact of face masks on speech intelligibility in individuals with varying degrees of hearing loss with special focus on acoustic and visual cues. The topic is relevant and may be interesting to the audience of this journal. However, there are a few major issues that should be addressed before it may be considered further for publication.
1. The Introduction section of this paper is particularly short. A more comprehensive review of previous literature is necessary to identify research gaps in the field.
2. This study is not well motivated. The authors did not explain in the Introduction section how their study goes beyond previous ones. For example, in Lines 55-57, they mentioned that “Atcherson, Mendel, Baltimore, Patro, Lee, Pousson and Spann [11] showed the greatest benefit of visual cues for speech intelligibility in individuals with severe hearing loss”. That finding is very similar to their hypothesis in Lines 67-68.
3. The experimental design of this study is not clearly described. For example, the determiner “the” should be used to refer to something that has been previously mentioned or is already known to readers. However, in Lines 125-126, the authors used “the video material” and “the speaker” out of the blue. Readers may get confused about them. As another example, in Line 132, the authors used the phrase “each list”, which is also confusing. What are the lists for? How many lists are there? I suggest the authors describe the experimental design more clearly. Also, there is a typo in the phrase “the extend of attenuation” in Line 128. It should be “the extent of attenuation”.
4. The authors mentioned that participants completed two audiovisual training lists prior to the actual measurements (Lines 139-140), but did not say why the training was necessary.
5. The authors should consider presenting their results in a clearer way. For example, in Lines 188-221, the authors used two paragraphs to report the means and SDs of SRTs and SRT differences. It would be much clearer to just present two tables.
6. The analysis method in this study is relatively simple. Most tests are monofactorial.
7. I’m not convinced by the authors’ interpretations of some of their results. For example, they did not find a direct correlation “between age and visual benefit (Line 355)”, which, according to them, supports the claim that “age has an indirect impact on audiovisual integration”. The authors may use that claim as a potential explanation for why the correlation was not significant, but I don’t think the insignificant result “supports” that claim.
8. The authors concluded that “individuals with poorer speech perception performance are more reliant on visual cues in noisy environments than less impaired listeners (Lines 400-401)”. The conclusion was reached based on the result of a correlation test reported in Lines 234-237. However, the p value of that test result is near the threshold (p = 0.02). Since multiple correlation tests were conducted in this study, this p value would actually not reach the significance threshold if it were corrected for multiple tests. In other words, the result that should support their main finding is not robust.
9. A writing problem of this paper is that it contained many very short, even one-sentence paragraphs, a practice that is very uncommon in academic writing. For example, there is only one sentence in the paragraphs in Lines 96-98 and 136-138. I suggest that the authors combine most of the short and related paragraphs into longer ones.
Comments on the Quality of English LanguageA writing problem of this paper is that it contained many very short, even one-sentence paragraphs, a practice that is very uncommon in academic writing. For example, there is only one sentence in the paragraphs in Lines 96-98 and 136-138. I suggest that the authors combine most of the short and related paragraphs into longer ones.
Author Response
Comment 1: The Introduction section of this paper is particularly short. A more comprehensive review of previous literature is necessary to identify research gaps in the field.
Response 1: Thank you for your comment. We have revised the introduction to better clarify our motivation for the study and hopefully improve the overview of the literature to date according to our research question (page 3, line 31).
Comment 2: This study is not well motivated. The authors did not explain in the Introduction section how their study goes beyond previous ones. For example, in Lines 55-57, they mentioned that “Atcherson, Mendel, Baltimore, Patro, Lee, Pousson and Spann [11] showed the greatest benefit of visual cues for speech intelligibility in individuals with severe hearing loss”. That finding is very similar to their hypothesis in Lines 67-68.
Response 2: In our opinion, previous literature has not clearly shown that the degree of hearing loss affects the amount of visual benefit differently. Our aim was to systematically compare groups with different degrees of hearing loss, reflecting a wide range of hearing impairments. We have revised the introduction to make the motivation for our study clearer (page 3, line 31).
Comment 3: The experimental design of this study is not clearly described. For example, the determiner “the” should be used to refer to something that has been previously mentioned or is already known to readers. However, in Lines 125-126, the authors used “the video material” and “the speaker” out of the blue. Readers may get confused about them. As another example, in Line 132, the authors used the phrase “each list”, which is also confusing. What are the lists for? How many lists are there? I suggest the authors describe the experimental design more clearly. Also, there is a typo in the phrase “the extend of attenuation” in Line 128. It should be “the extent of attenuation”.
Response 3: We revised and modified the whole Materials and Methods section in order to improve clarity (page 4, line 88). ‘List’ refers to test lists of 20 sentences used to determine the SRT80 in each condition (or percent correct speech intelligibility in case of the visual-only condition).
Comment 4: The authors mentioned that participants completed two audiovisual training lists prior to the actual measurements (Lines 139-140), but did not say why the training was necessary.
Response 4: Two audio-visual test lists were completed prior to the actual measurements to reduce training effects, as suggested by previous studies using the same test material. This is now reported in the methods section (page 6, line 156).
Comment 5: The authors should consider presenting their results in a clearer way. For example, in Lines 188-221, the authors used two paragraphs to report the means and SDs of SRTs and SRT differences. It would be much clearer to just present two tables.
Response 5: Most of the results (medians and interquartile ranges) are now presented in a table, and a more detailed description of the results is now in the text (page 8, table 2). We decided to change from means and SDs to median and interquartile ranges because we changed the statistical analysis to use only non-parametric tests due to the small and variable group sizes (as criticized by another reviewer).
Comment 6: The analysis method in this study is relatively simple. Most tests are monofactorial.
Response 6: We appreciate the comment. The use of relatively simple analysis methods, including monofactorial tests, is a valid approach in this study, particularly given the small dataset and focused objectives. Monofactorial tests are well-suited for isolating individual effects, providing clear and interpretable results. The small sample size further justifies this choice, as more complex multifactorial analyses could risk overfitting or unreliable outcomes. However, in further studies with greater cohorts multivariable analysis should be included to better estimate the influence of clinical factors (e.g., PTA4, speechreading score) on parameters such as visual benefit.
Comment 7: I’m not convinced by the authors’ interpretations of some of their results. For example, they did not find a direct correlation “between age and visual benefit (Line 355)”, which, according to them, supports the claim that “age has an indirect impact on audiovisual integration”. The authors may use that claim as a potential explanation for why the correlation was not significant, but I don’t think the insignificant result “supports” that claim.
Response 7: Thank you for your feedback. We apologize for the misunderstanding created by our phrasing. We have revised the section to ensure our interpretation is clearer and more accurately reflects the results (page 12, line 373). We appreciate your thoughtful comment, which has helped us improve the clarity of our discussion.
Comment 8: The authors concluded that “individuals with poorer speech perception performance are more reliant on visual cues in noisy environments than less impaired listeners (Lines 400-401)”. The conclusion was reached based on the result of a correlation test reported in Lines 234-237. However, the p value of that test result is near the threshold (p = 0.02). Since multiple correlation tests were conducted in this study, this p value would actually not reach the significance threshold if it were corrected for multiple tests. In other words, the result that should support their main finding is not robust.
Response 8: We appreciate the reviewer’s comment and agree that the interpretation of this result was questionable. As visual benefit is nested within the audio-only results, we have removed this correlation analysis from the manuscript. We have replaced it with a correlation analysis of best-aided PTA4 with visual benefit and revised the results accordingly (page 10, line 263). Thank you for highlighting this issue.
Comment 9: A writing problem of this paper is that it contained many very short, even one-sentence paragraphs, a practice that is very uncommon in academic writing. For example, there is only one sentence in the paragraphs in Lines 96-98 and 136-138. I suggest that the authors combine most of the short and related paragraphs into longer ones.
Response 9: Thank you for pointing this out. We revised the manuscript extensively and tried to avoid using small paragraphs.
Reviewer 3 Report
Comments and Suggestions for Authors
The manuscript addresses the impact of masks on speech intelligibility, examining both sound attenuation and the reduction of visual cues. Specifically, the analysis focuses on comparing individuals with varying degrees of hearing loss to a control group with normal hearing. The hypothesis is that individuals with severe hearing loss will show a greater visual benefit and will therefore be more affected by the face masks than NH listeners.
*Participants
NH(n=16); MoHL(n=11); MiHL(n=13); SHL(n=10). It is an important concern that the n of each group are different. The authors did not provide sufficient support when they decided to conduct their statistical analyses. In this regard they only mention that they performed a t-test, ANOVA or correlation, but given the difference in number of participants per group it is not clear whether they performed an analysis of variance to support their decision. Depending on that result, Welch's t-test and ANOVA may have been the best choice.
The ages of the participants are significantly different. This is another important problem. Furthermore, as mentioned by the authors, no assessment was made of possible cognitive problems and cognitive reserve of the participants and their levels of vision loss. So how can one be sure that the results obtained are not influenced by this variable (age)?
Participants were tested with hearing aids: mild hearing loss (MiHL, 2 participants), moderate hearing loss (MoHL, 6 participants), and severe hearing loss (SHL, 10 participants). This raises several questions:
- Does the level of hearing loss change when a person uses a hearing aid?
- How does the hearing aid affect cognitive, auditory, and visual performance?
- How does the variable 'years of hearing aid use' impact cognitive, auditory, and visual performance?
- How does the type of device used affect performance?
- How does the relationship between the duration of hearing loss and the duration of hearing aid use affect performance?
Again, it is not possible to know whether the results obtained are influenced by these variables or not.
* Experimental Design
It is mentioned that the procedure of [10] was followed. There it is explained that the noise was presented at a fixed SPL of 65 dB and that the speech signal was varied from SNR = -5 to a maximum difference of 20 dB (speech at 85 dB SPL). The problem arises because this design reproduces a stimulus on each trial at a varying (usually increasing) SPL. This is a confounding effect as participants may perceive differences in the overall intensity of the sound and not only in the SNR. This complicates the interpretation of the results.
The visual benefits were only assessed in the unmasked case. Does this mean that the visual benefits can be applied linearly to the masked cases? For example, if the visual benefits are -3 dB (SRT difference), does that mean that if my SRT is -1 dB with the surgical mask my intelligibility would increase to -4 dB if I had the visual benefits? Hard to believe because it was not evaluated (AV condition, but with sound attenuation due to the type of mask).
* Results
In SRT and Speechreading, values are mentioned, but no statistical analysis of significant relationships. The high variance in the HI groups in relation to the NH group is striking.
When describing the attenuation results by group, it was unclear what the authors were referring to. For clarity, significant differences should be marked on the figure. The authors mention significant differences for all groups in the FFP2 mask condition; however, this is difficult to believe given that the means for each group are nearly identical in Figure 3. A similar issue arises with the results for the surgical mask and visual benefits. Although t-tests are mentioned, it is not specified which pairs of groups were analyzed, contributing to the confusion.
A 4 (groups) x 3 (partial mask impact) ANOVA was performed, but it is unclear why the three cases were combined as ‘partial mask impact,’ given that one pertains to visual benefits and the other two to sound attenuation. For example, why was it relevant to determine whether the visual benefits for NHs were significantly different from the FFP2 attenuation effects for MoHLs? Without further explanation it seems totally irrelevant.
A moderate relational analysis is presented, but was it not possible to conduct an analysis to determine whether there is a significant difference between the AO case and visual benefits across the HI groups? Given the available data, it seems feasible; why was this not done, and why was a correlational analysis preferred instead?
Results do not follow APA format.
For all the above reasons I am not convinced of the results. There are several confounding variables and the methodology does not address the problem adequately.
Author Response
Comment 1: NH(n=16); MoHL(n=11); MiHL(n=13); SHL(n=10). It is an important concern that the n of each group are different. The authors did not provide sufficient support when they decided to conduct their statistical analyses. In this regard they only mention that they performed a t-test, ANOVA or correlation, but given the difference in number of participants per group it is not clear whether they performed an analysis of variance to support their decision. Depending on that result, Welch's t-test and ANOVA may have been the best choice.
Response 1: We have changed the statistical analyses to include only non-parametric tests, which are more robust to small and varying group sizes.
Comment 2: The ages of the participants are significantly different. This is another important problem. Furthermore, as mentioned by the authors, no assessment was made of possible cognitive problems and cognitive reserve of the participants and their levels of vision loss. So how can one be sure that the results obtained are not influenced by this variable (age)?
Response 2: This is a limitation of our study, although the only difference in age was that the normal hearing group was younger (but still elderly) than the moderate and severe hearing groups. We have discussed this issue in more detail (page 12, line 388).
Comment 3: Participants were tested with hearing aids: mild hearing loss (MiHL, 2 participants), moderate hearing loss (MoHL, 6 participants), and severe hearing loss (SHL, 10 participants). This raises several questions:
- Does the level of hearing loss change when a person uses a hearing aid?
- How does the hearing aid affect cognitive, auditory, and visual performance?
- How does the variable 'years of hearing aid use' impact cognitive, auditory, and visual performance?
- How does the type of device used affect performance?
- How does the relationship between the duration of hearing loss and the duration of hearing aid use affect performance?
Again, it is not possible to know whether the results obtained are influenced by these variables or not.
Response 3: We understand the reviewer's concerns. We have tried to present the situation as realistically as possible to reflect the fact that people with mild to moderate hearing loss often do not use hearing aids in everyday life. We added a correlation analysis between visual benefit and hearing level at test ('real' hearing level) to reflect the hearing thresholds available at test. We also looked at the effect of years of hearing aid use but found no effect on visual benefit. However, it is often difficult to interpret the time information as it is often not very reliable due to mainly slowly progressing hearing loss. The same is true for the time of onset of hearing loss. We have revised the discussion to better highlight the possible influence of hearing aid use and its possible interaction with cognition (page 12, line 377). Participants with other hearing (e.g., cochlear implants) devices than hearing aids were not included in this study.
* Experimental Design
Comment 4: It is mentioned that the procedure of [10] was followed. There it is explained that the noise was presented at a fixed SPL of 65 dB and that the speech signal was varied from SNR = -5 to a maximum difference of 20 dB (speech at 85 dB SPL). The problem arises because this design reproduces a stimulus on each trial at a varying (usually increasing) SPL. This is a confounding effect as participants may perceive differences in the overall intensity of the sound and not only in the SNR. This complicates the interpretation of the results.
Response 4: The mentioned SNR of -5 dB was the starting SNR for the adaptive procedure, where the speech level was adjusted according to the listener's response, resulting in an intelligibility of 80%. We hope this is now clearer in the modified methods section (page 6, line 152).
Comment 5: The visual benefits were only assessed in the unmasked case. Does this mean that the visual benefits can be applied linearly to the masked cases? For example, if the visual benefits are -3 dB (SRT difference), does that mean that if my SRT is -1 dB with the surgical mask my intelligibility would increase to -4 dB if I had the visual benefits? Hard to believe because it was not evaluated (AV condition, but with sound attenuation due to the type of mask).
Response 5: When masked, the visual benefit is lost and there may be an additional deterioration in speech intelligibility due to the acoustic attenuation effects, which are greater for FFP2 masks than for surgical masks and more pronounced in the higher frequency ranges. We assessed visual benefit by calculation:
visual benefit = (audiovisual condition) – (audio-only condition).
The acoustic attenuation was assessed by calculation as well:
Acoustic attenuation surgical mask = (audiovisual with surgical mask) - (audio-only)
Acoustic attenuation FFP2 mask = (audiovisual with FFP2) – (audio-only)
Thus, the loss of visual cues (= visual benefit) and the effect of acoustic attenuation add up to the overall effect of a mask.
For example, if the visual benefits are -3 dB (SRT difference), does that mean that if my SRT is -1 dB with the surgical mask my intelligibility would increase to -4 dB if I had the visual benefits?
Yes, we would expect that and even a slight improvement when presenting the unattenuated audio signal. We did not measure SRT with mask but with unattenuated audio because we expected the same result as in the audio-only condition.
* Results
Comment 6: In SRT and Speechreading, values are mentioned, but no statistical analysis of significant relationships. The high variance in the HI groups in relation to the NH group is striking.
Response 6: We show absolute SRT values for all groups in figure 2 (page 7) with adequate analysis which is shown in the figure and described in the text. Due to the range of hearing loss severity in each group variability is expected to be higher compared to NH. Speechreading scores are shown in figure 4 (page 9) and results are described in the text, accordingly.
Comment 7: When describing the attenuation results by group, it was unclear what the authors were referring to. For clarity, significant differences should be marked on the figure. The authors mention significant differences for all groups in the FFP2 mask condition; however, this is difficult to believe given that the means for each group are nearly identical in Figure 3. A similar issue arises with the results for the surgical mask and visual benefits. Although t-tests are mentioned, it is not specified which pairs of groups were analyzed, contributing to the confusion.
Response 7: We added significances to the figure. We found significant effects between the audio-visual condition and, for example, the FFP2-mask condition in each hearing status groups. The statistical analysis was done separately for each hearing status group. We also looked at the amount of improvement, when visual information was available compared to the audio-only situation and found no difference between the groups (see figure 3, page 8). The same was done for the attenuation effects of the two masks, where we also found no difference between the groups (see also figure 3). Since we have extensively revised the results and discussion sections, we hope this now clearer.
Comment 8: A 4 (groups) x 3 (partial mask impact) ANOVA was performed, but it is unclear why the three cases were combined as ‘partial mask impact,’ given that one pertains to visual benefits and the other two to sound attenuation. For example, why was it relevant to determine whether the visual benefits for NHs were significantly different from the FFP2 attenuation effects for MoHLs? Without further explanation it seems totally irrelevant.
Response 8: We agree that comparing visual benefit and acoustic attenuation within the same group is irrelevant. We modified the statistical analysis and performed separate Kruskal-Wallis tests on the visual benefit and the two attenuation characteristics to analyze whether the visual benefit or the acoustic attenuations differed between the different hearing status groups (page 8, line 237). Friedman test was used to make relevant comparisons for each group between conditions (page 7, line 198).
Comment 9: A moderate relational analysis is presented, but was it not possible to conduct an analysis to determine whether there is a significant difference between the AO case and visual benefits across the HI groups? Given the available data, it seems feasible; why was this not done, and why was a correlational analysis preferred instead?
Response 9: Comparison of auditory outcomes (AO) between groups is not directly feasible due to the grouping based on PTA4, which is strongly linked to AO. However, the comparison of visual benefits between groups is a key feature of our study, as presented in figure 3. We preferred a correlation analysis to allow for a continuous analysis of hearing status, as opposed to fixed groups defined by WHO criteria.
Comment 10: Results do not follow APA format.
Response 10: The results section has been revised accordingly.
Reviewer 4 Report
Comments and Suggestions for Authors
This study delved into the effects of face masks on speech intelligibility among individuals with normal hearing (NH) and varying degrees of hearing impairment (HI). Although the investigation presents intriguing insights, several key questions warrant further clarification:
Experiment Setup and Protocol: The experimental setup and protocol description is rather succinct. A comprehensive account is necessary to facilitate replication and understanding of the study's methodology.
Audio Signal Editing: How were the audio signals modified to accurately represent the acoustic attenuation properties of each mask type? A detailed explanation of the procedures, including any filters or techniques employed, should be provided.
Background Sound Pressure Level: The background sound pressure level was set at 65 dB. What was the rationale behind this choice? Additionally, was this level A-weighted? Further, what was the signal-to-noise ratio employed in the study?
Audio-Only and Audiovisual Conditions: Did both the audio-only and audiovisual conditions incorporate audios attenuated by the presence of masks?
Audiovisual Conditions with Masks: In the conditions involving audiovisual input with a surgical mask and FFP2 mask, did the study include both attenuated and unattenuated audio recordings?
Results Figures: The figures presenting the results should explicitly indicate significant symbols to enhance clarity and comprehension.
Author Response
Comment 1: Experiment Setup and Protocol: The experimental setup and protocol description is rather succinct. A comprehensive account is necessary to facilitate replication and understanding of the study's methodology.
Response 1: We have extensively revised the entire methods section (page 4, line 88). We hope that the setup and experimental protocol are now more clearly described.
Comment 2: Audio Signal Editing: How were the audio signals modified to accurately represent the acoustic attenuation properties of each mask type? A detailed explanation of the procedures, including any filters or techniques employed, should be provided.
Response 2: We now describe the modification of the audio signal according to the masks in more detail (page 5, line 140). A figure of the attenuation properties can be found in Sönnichsen et al. 2022b. We refer to this publication, which is freely available, since we used the same experimental setup.
Comment 3: Background Sound Pressure Level: The background sound pressure level was set at 65 dB. What was the rationale behind this choice? Additionally, was this level A-weighted? Further, what was the signal-to-noise ratio employed in the study?
Response 3: We used 65 dB SPL as the background noise level because it is widely used in audiological practice and reflects normal conversation volume. C-weighing was used for calibration to avoid the influence of low-frequency noise on the level. We did not measure at fixed SNRs to avoid ceiling effects, especially in the normal hearing and mild hearing loss groups. Instead, we determined the speech reception threshold at 80% intelligibility for each subject using an adaptive procedure. We have revised the methods section accordingly to make this clearer (page 5, line 184).
Comment 4: Audio-Only and Audiovisual Conditions: Did both the audio-only and audiovisual conditions incorporate audios attenuated by the presence of masks?
Response 4: The audio signal included mask attenuation only when the mouth and nose were covered by a mask-like object in the video. Otherwise (audiovisual and audio-only condition), the audio signals were unattenuated (original signals from the Female German Matrix Test). We hope that the changes in the methods section now make this clear.
Comment 5: Audiovisual Conditions with Masks: In the conditions involving audiovisual input with a surgical mask and FFP2 mask, did the study include both attenuated and unattenuated audio recordings?
Response 5: In the masked conditions, only attenuated audio was used. The attenuation was applied according to the respective mask. We would expect the same result as in the audio-only condition if we had measured unattenuated speech in combination with the visual information provided in the masked condition (mouth and nose covered by a mask-shaped object).
Comment 6: Results Figures: The figures presenting the results should explicitly indicate significant symbols to enhance clarity and comprehension.
Response 6: Thank you for this valuable comment. In addition to the colors, different symbols were assigned to the different hearing status groups. We replaced Figure 2, where we originally only distinguished between NH and HI (together in one group), with a figure showing the SRTs of each condition for each hearing status group. We believe that this change better reflects the aim and design of the study.
Round 2
Reviewer 3 Report
Comments and Suggestions for Authors
Recommended for publication.
Author Response
Comment 1: Recommended for publication.
Response 1: Thank you for your valuable comments.
Reviewer 4 Report
Comments and Suggestions for Authors
The authors have addressed the initial queries, yet a new concern emerges regarding confirming the effects of face masks on speech intelligibility: whether these effects are primarily due to visual or acoustic factors. While the authors have incorporated a visual-only condition into their study, the conclusion requires further substantiation through data derived from unattenuated signals with masks without visual clues. This necessity arises from the observation that listeners exhibit markedly different performances when exposed to attenuated versus unattenuated signals, irrespective of the presence or absence of visual cues, as previously demonstrated.
Author Response
Comment 1:
he authors have addressed the initial queries, yet a new concern emerges regarding confirming the effects of face masks on speech intelligibility: whether these effects are primarily due to visual or acoustic factors. While the authors have incorporated a visual-only condition into their study, the conclusion requires further substantiation through data derived from unattenuated signals with masks without visual clues. This necessity arises from the observation that listeners exhibit markedly different performances when exposed to attenuated versus unattenuated signals, irrespective of the presence or absence of visual cues, as previously demonstrated.
Response 1:
We sincerely thank the reviewer for their valuable comment. In our initial investigations into the effects of face masks, we included a condition featuring a face mask paired with unattenuated audio in a study conducted with individuals with normal hearing. However, that study did not reveal a significant difference between the aforementioned condition and the audio-only condition (Sönnichsen et al., Otol Neurotol, 2022). Based on these findings, we opted not to include a masked condition with unattenuated audio in the present study.
Round 3
Reviewer 4 Report
Comments and Suggestions for Authors
I have reservations about the response. It is crucial to recognize that findings derived from populations with normal hearing may not necessarily apply to individuals with hearing loss. The absence of significant effects in one group does not inherently confirm the same outcome in another.
Author Response
Comment 1: I have reservations about the response. It is crucial to recognize that findings derived from populations with normal hearing may not necessarily apply to individuals with hearing loss. The absence of significant effects in one group does not inherently confirm the same outcome in another. Answer 1: Thank you for your comment regarding the generalizability of our findings in normal-hearing individuals. We agree that results obtained from participants with normal hearing cannot be directly extrapolated to individuals with hearing impairment. While potential differences between these populations may be modest, we acknowledge this methodological consideration. We have addressed this aspect in our limitations section (lines 395-400), emphasizing the need for cautious interpretation when extending these findings to hearing-impaired populations. We appreciate your attention to this point, as it strengthens our manuscript.